# The Role of P16, P53, KI-67 and PD-L1 Immunostaining in Primary Vaginal Cancer

**DOI:** 10.3390/cancers15041046

**Published:** 2023-02-07

**Authors:** Eva K. Egger, Mateja Condic, Damian J. Ralser, Milka Marinova, Alexander Mustea, Florian Recker, Glen Kristiansen, Thore Thiesler

**Affiliations:** 1Department of Gynecology and Gynecological Oncology, University Hospital, 53127 Bonn, Germany; 2Department of Nuclear Medicine, University Hospital, 53127 Bonn, Germany; 3Department of Pathology, University Hospital, 53127 Bonn, Germany

**Keywords:** vaginal cancer, immunotherapy, anti-VEGF-therapy

## Abstract

**Simple Summary:**

To date, vaginal cancer is the only tumor entity of the female genital tract without a practical guideline within the National Comprehensive Cancer Network (NCCN). Therapeutic options vary between surgery for Stage I disease and concurrent chemoradiation for Stage II to IV disease. The lack of data regarding systemic therapies remains challenging to overcome as vaginal cancer is too rare to conduct large, randomized trials. By assessing pathological and immunhistochemical variables in this rare tumor entity, as well as the clinical courses of patients treated within different treatment schedules including immunotherapy and anti-VEGF-therapy, we aimed to show similarities to cervical cancer where emerging therapeutic options have improved survival significantly.

**Abstract:**

Background: To analyze clinical, pathological and immunohistochemical correlates of survival in vaginal cancer patients. Methods: Retrospective analysis of primary vaginal cancer patients, treated at the Department of Gynecology and Gynecological Oncology of the University Hospital Bonn between 2007 and 2021. Results: The study cohort comprised 22 patients. The median age was 63 years (range: 32–87 years). Squamous cell histology was present in 20 patients. Five-year OS in Stage I, II, III and IV was 100%, 56.25%, 0% and 41.67%, respectively (*p* = 0.147). Five-year DFS was 100%, 50%, 0% and 20.83%, respectively (*p* = 0.223). The 5-year OS was significantly reduced in the presence of nodal metastasis (*p* = 0.004), lymphangiosis (*p* = 0.009), hemangiosis (*p* = 0.002) and an age above 64 years (*p* = 0.029). Positive p 16 staining was associated with significantly improved OS (*p* = 0.010). Tumoral and immune cell PD-L1 staining was positive in 19 and in 16 patients, respectively, without significant impact on OS; 2 patients with metastastic disease are long-term survivors treated with either bevacizumab or pembrolizumab. Conclusion: P16 expression, absence of lymph- or hemangiosis, nodal negative disease and an age below 64 years show improved survival rates in PVC. Tumoral PD-L1 expression as well as PD-L1 expression on immune cells is frequent in PVC, without impacting survival. Within our study cohort, long-term survivors with recurrent PVC are treated with anti-VEGF and immunotherapy.

## 1. Introduction

To date, vaginal cancer accounts for only 0.1% of all malignancies [1]. Its estimated incidence is 17,600 cases and about 8000 deaths annually worldwide [1]. Primary vaginal cancer (PVC) is only considered in the absence of vulvar or cervical cancer or their local recurrences [2]. Approximately 80 to 90% of PVC exhibit squamous cell histology and 4 to 10% are adenocarcinomas [2]. Squamous cell PVC is often characterized by a persistent human papilloma virus (HPV) infection, predominantly with HPV type 16 [2]. Vaginal adenocarcinomas are frequently associated with an intrauterine exposition to Diethylstilbestrol (DES). Adenocarcinomas arising independently of DES exposition show an especially poor prognosis [3,4]. Due to the rarity of PVC, current treatment recommendations remain controversial between concurrent radiochemotherapy and surgery [2,5]. Recurrence rates range according to the disease stage between 24% in Stage I and 83% in Stage IV disease. In case of recurrence, further treatment recommendations are scarce. As about 50% to 80% of PVCs seem to be HPV-dependent, there may be a biological resemblance to cervical cancer where emerging data show the benefit of immunotherapy and bevacizumab [6,7]. Ribonucleotid reductase (RNR) overactivity, frequently seen in cervical cancer patients, promotes DNA damage repair and leads to tumor cell survival. In a phase II trial also including four PVC patients, the combination therapy of triapine, an RNR inhibitor, and cisplatin-radiotherapy, led in two patients to a survival benefit compared to cisplatin-radiotherapy only in the two other patients, also implicating a similar biological behavior of PVCs and cervical cancer [8]. As the impaired virus clearance by the immune system is the carcinogenesis driver of HPV-dependent tumors, immunotherapy seems very promising in these tumors. So far, the Checkmate 358 trial included two PVC patients without response to nivolumab [9]. Another basket trial showed response to pembrolizumab monotherapy in one out of two PVC patients [10]. Currently, there are two trials including vaginal cancer patients which have finished their actual recruitment: the Phase I SABR-Trial for rare tumors, with durvalumab, tremelilumab and pelvic radiation (NCT03277482) and the study of the National Cancer Institute analyzing adavosertib, a wee-1 Inhibitor, cisplatin and radiation therapy for cervical, upper vaginal and uterine cancers (NCT03345784). Another phase Ib/II trial evaluated avelumab in patients with HPV-16 positive recurrent or metastatic malignancies, including five patients with vulvar or vaginal cancer. Avelumab was given in combination with TG4001, an HPV E6/E7 vaccine, leading to one complete response and seven partial responses in the entire cohort [11].

Here, we analyzed the immunohistochemical profile and the clinical outcome in a series of PVC patients treated across various therapeutic lines, including therapies from the therapeutic spectrum of cervical cancer patients.

## 2. Material and Methods

### 2.1. Data Collection

This study was conducted according to the guidelines of the Declaration of Helsinki and approved by the ethics committee of the Faculty of Medicine at the University of Bonn, Germany (Nr: 328/22). The institutional record database was screened for vaginal cancer patients treated at the Department of Gynecology and Gynecological Oncology between January 2010 and December 2021. Tissue collection was conducted within the Biobank initiative of the University of Bonn. The only inclusion criterion was a histologically confirmed diagnosis of a primary vaginal cancer. The only exclusion criterion was a history of vulvar or cervical cancer. All patients provided written informed consent before tissue collection. Baseline characteristics, pathology and therapeutic course were recorded from patient’s charts, surgery reports, radiation protocols and pathologic findings. Follow-up data were updated in July 2022. Histopathological diagnosis was determined based on World Health Organization (WHO) criteria, considering only patients with no prior history of a cervical or vulvar cancer [12]. Tumor stage was based on the 2018 revised International Federation of Gynecology and Obstetrics (FIGO) system and the TNM-Classification of the Union for International Cancer Control (UICC) [13,14].

### 2.2. Patients

There were 26 patients in total; 4 patients were excluded from further analysis (one patient was lost to follow-up; for 3 patients, no tumor material was available for immunohistochemical staining). Immunohistochemical evaluation was performed by a pathologist with focus on gynecologic pathology (T.T.). Immunohistochemical evaluation was performed based on tissue microarrays of 17 patients and based on whole-tumor slides of 5 patients.

### 2.3. Tissue Microarray (TMA) Creation and Immunhistochemistry

Tissue micro arrays were prepared by the Institute of Pathology of the University of Bonn. The hematoxylin and eosin (HE) Slides of all 22 patients were examined by a pathologist with focus on gynecologic pathology (Thore Thiesler) to confirm the diagnosis. In all patients, 2 representative tumor areas were identified and marked on 1 slide stained with HE. From the correlating formalin fixed paraffin-embedded tissue block (FFPE), 1 mm core biopsies (0.875 mm^2^) were taken from the identified tumor nests. In each case, 2 core biopsies were taken from each paraffin block to avoid tumor heterogeneity. For each patient, there were 2 TMAs with 16 to 32 samples. The following antibodies were used for immunohistochemistry: rabbit anti-human PD-L1 IgG monoclonal antibody (clone ZR3, dilution 1:50; Zeta Corporation, Arcadia, CA, USA), CINtec^©^ Histology kit for the evaluation of p16^INK4a^ (Roche, Basel, Switzerland), mouse anti-human p53 IgG2b monoclonal antibody (clone DO-7, dilution 1:500; Dako, Glostrup, Germany) and mouse anti-human Ki-67 IgG1 monoclonal antibody (clone MIB-1, dilution 1:500; Dako).

Immunostaining on TMAs and on whole slides was performed for PD-L1, p16, p53 and Ki-67, applying an automated staining system. Immunhistochemical staining was performed on a Ventana Benchmark system (BenchMark ULTRA; Ventana Medical Systems, Tucson, AZ, USA) for p16 and on a Medac 480S system (Medac GmbH, Wedel, Germany) for PD-L1, p53 and Ki-67 using established staining protocols of the routine laboratory. An UltraView Universal DAB Detection Kit (Ventana Medical Systems Inc., Tucson, AZ, USA) was used on the Ventana Benchmark system and the HRP colour-coded BrightVision (Immunologic WellMed B.V., Duiven, The Netherlands) detection system on the Medac system.

### 2.4. Evaluation of Immunohistochemistry (IHC)

Tumor tissue presence was validated on the HE-stained slides and the HE-stained TMA slides by visual examination. P53, p16, Ki-67, PD-L1 combined positivity score (CPS-score), PD-L1 immune cell score (IC-Score) and PD-L1 tumor proportional score (TPS) were evaluated. P53 was evaluated according to a recently published protocol with 6 major p53 IHC patterns [15]. Positive P16 staining was considered in case of a diffuse or strong staining of the basal and/or parabasal cells irrespective of staining of the superficial cell layers [16]. Ki-67 was considered as positive in case of a nuclear staining of cancer cells, and the percentage of stained cells was recorded. PD-L1 staining was considered as present in case of a membranous staining of tumor cells and immune cells. Immune cells were only considered as positive if present within the same high resolution visual field (40×) irrespective of the staining intensity and including all viable tumor cells on the slide. Tumor cells were scored by the TPS-score, immune cells by the IC-score and both by the CPS-score [17].

### 2.5. Statistical Analysis

Statistical analyses were performed using Minitab Version 18, Minitab LLC., State College, PA, USA. The survival analyses for progression-free survival (PFS) and overall survival (OS) are based on the Kaplan–Meier method. The time-to-event intervals were described in months from the date of primary diagnosis until the date of the event. The data were censored at the date of the last follow-up if there was not an event. Using the log-rank test, 5-year-survival-curves were compared on a 95% confidence level. Identified significant factors were analyzed by multivariate regression analysis. Clinicopathological factors and immunohistochemistry were correlated by Fisher’s exact test.

## 3. Results

### 3.1. General Patient Characteristics

The study cohort comprised 22 patients. A total of 9 patients died during the follow-up period; 12 patients experienced a relapse, of whom 3 were still alive as of this writing. Median follow-up was 18 months (range 3–156 months). The median age was 63 years (range: 32–87 years). There were 2 adenocarcinomas and 20 squamous cell carcinomas. Six primary tumors were in the lower vaginal third, two were located in the middle vaginal third, six were located in the upper vaginal, and eight patients had a tumor affecting the whole vagina. The median tumor diameter was 4 cm (range 1.1–9 cm). FIGO stages were distributed as follows: I: five patients, II: eight patients, III: three patients, IVA: one patient, IVB: five patients. A total of 18 patients received different kinds of surgery in first line; 3 patients received concurrent radiochemotherapy, and 1 patient received only systemic therapies. Tumor-free margins in final pathology were present in 14 out 18 patients treated by surgery. The median depth of infiltration was 9 mm (range 1–38 mm). Seven patients showed a lymphangiosis, and four patients showed a hemangiosis in the final histology. Surgery comprised nine primary and one secondary exenteration (six total/four anterior), two radical hysterectomies with partial colpectomy, two partial colpectomies, three total radical colpectomies, and one patient received a palliative entero- and urostomy only. Further details for therapeutic interventions are depicted in Table 1. Patients 13, 17 and 19 are long-time survivors after recurrence. Patient 13 had received a complete exenteration. Inguinal lymph node dissection and radiation of the inguinal and pelvic region was performed after inguinofemoral recurrence. The second recurrence occurred within the ileocecal region. After resection of the ileocecal colon, she received carboplatin/paclitaxel/bevacizumab and had been on bevacizumab for 46 months as of this writing with no evidence of disease at follow-up. Patient 17 had received a radical colpectomy for a 1.7 cm tumor. She recurred after 96 months and was under palliation only at the date of data recording. Patient 19 had an adenocarcinoma with pulmonal metastasis at first diagnosis and received only systemic therapies including bevacizumab. She had been on pembrolizumab only for 31 months as of this writing. A detailed therapy course of this patient has been previously published elsewhere [18]. All three patients in FIGO Stage III died. Patient 15 developed an inguinal recurrence and pulmonal metastasis and finally died in the following course. Patient 21 died due to a severe infection after palliative arterial chemoperfusion. Patient 22 received an arterial embolization due to massive tumor bleeding and declined further therapy afterwards and died 2 months later. The therapeutic course, histology and FIGO stage of all 22 patients are depicted in Table 1. Patients with recurrence are marked in light grey.

### 3.2. 5-Year-Survival Data

Survival analysis showed significant decreased OS and DFS in case of nodal metastasis, lymphangiosis and hemangiosis, a missing p16 expression and an age above 64 years. The depth of tumor infiltration was only relevant regarding DFS but not OS. No other factors showed significance regarding survival as depicted in Table 2. A positive p16 expression showed a significant association with a younger age (*p* = 0.0351) and a depth of infiltration less than 7 mm (*p* = 0.0379). In the multivariate analysis, none of the five factors (age, lymphangiosis, hemangiosis, nodal metastases and missing p16 immunostaining) remained significant regarding survival as depicted in Table 3.

The correlation of immunohistochemistry and clinicopathological factors showed only a significant correlation of a PDL1-CPS > 1 in case of a tumor size > 4 cm and in case of p53 wildtype and L0 and V0 as depicted in Table 4.

In 19 patients, there was a positive staining for PD-L1 on the tumor cells (PD-L1 TPS score range 0–68). Regarding the immune cells, positive staining was present in 16 patients only (PD-L1 IC score range: 1–5). The PD-L1CPS score was positive in 19 patients (range: 0–51). Intratumoral T-cell infiltration was present in 10 patients and absent in 7 patients. Four out of these seven patients showed no PD-L1 expression on the immune cells, however, except for one patient, all were positive for PD-L1 on tumor cells. No survival differences were observed regarding PD-L1- CPS, PD-L-1 TPS and PD-L1 IC. Further details are depicted in Table 4. Examples of a membranous PD-L1 expression, a nuclear and cytoplasmatic *p* 16 expression, nuclear p 53 overexpression, strong nuclear and cytoplasmatic KI-67 staining in PVC are shown in Figure 1. None of the PD-L1-scores showed any association with survival as depicted in Table 5.

## 4. Discussion

To date, vaginal cancer is the only tumor entity of the female genital tract without a practical guideline within the National Comprehensive Cancer Network (NCCN). The German practical guideline recommends surgery for Stage I disease only. Stage II to IV disease should be treated by concurrent chemoradiation (CCRT). Only in Stage IV disease is pelvic exenteration recommended within an individual decision process [19]. Brachytherapy within the concept of concurrent chemoradiation shows a prolongation of survival of more than 2 years compared to external CCRT only [20]. Treatment recommendations for recurrent and metastatic disease are missing due to a lack of data regarding systemic therapy in this tumor entity [19]. Further, data to encircle the biologic behavior of this disease are missing. A recent review identified increasing tumor size, disease stage and the presence of nodal metastases to impact survival and concluded that surgery and primary concurrent radiotherapy seem to have equal results in Stage I and II disease, while Stage III and IV disease should be treated by brachytherapy and external beam radiation only [21].

In this study, we analyzed clinicopathological and immunohistochemical variables in PVC regarding their prognostic value and provide an overview of different therapeutic lines in PVC. Nodal metastases, lymphangiosis and hemangiosis, a negative p16 status and an age above 64 years showed significant prognostic values regarding OS. The depth of tumor infiltration was only relevant regarding DFS.

Accumulating evidence suggests an HPV-dependent and an HPV-independent pathway in the carcinogenesis of PVC [19]. HPV, especially HPV type 16 is present in about 50 to 80% of all PVCs. HPV positive PVCs show improved survival rates, especially in advanced disease stages compared to HPV negative PVCs [22,23,24,25]. As p16 immunostaining is positive in more than 97% of all HPV positive PVCs, it might serve as an easy and reliable surrogate marker for the distinction between these two tumor types. Most importantly, it is not affected by long storage times of FFPE tissues compared to HPV DNA detection. This, however, might be the reason for highly different HPV rates within different studies regarding PVC [22,23,24]. P16 negative PVC patients show impaired DFS and OS compared to p16 positive PVC patients with respect to definite radiotherapy [26]. Comparing the histological features, HPV-positive PVCs are often non-keratinizing, warty and basaloid-like compared to HPV negative PVCs with a predominantly keratinizing phenotype [27]. Similar findings were observed in cervical cancer where keratinization was associated with reduced radiosensitivity and a shorter OS. Interestingly, this survival difference was resolved in the case of surgically treated cervical cancer patients [28]. Considering primary treatment options of surgery or definite concurrent radiochemotherapy, p16 IHC might help in the decision-making process.

In our cohort, we found 5 patients with negative p16 immunohistochemistry. Of this subgroup, 3 patients died within 4 to 12 months after primary diagnosis. The remaining 2 patients were still alive at 4 and 12 months follow-up. None showed a tumor location in the upper third of the vagina. Hence, negative p16 immunostaining was associated with a significantly decreased OS.

Regarding lymphangiosis, hemangiosis and locoregional lymph node metastasis, no correlation with a negative p 16 immunostaining was seen in our cohort. Only age and depth of infiltration showed a significant association. As previously shown by others, age and locoregional lymph node metastases significantly decreased OS and DFS in our cohort [21]. While lymph- and hemangiosis are known risk factors for an impaired DFS and OS in cervical cancer, we did not find any published evidence for PVC [29]. In our cohort, both proved to negatively impact DFS and OS in PVC significantly.

Results on the predictive relevance of PD-L1 expression in cervical cancer are conflicting. PD-L1 expression in cervical cancer is not caused by gene amplification but by its oncogene E7 which is directly associated with a tumoral PD-L1 expression, leading to an impaired CD8+ T-cell function. Therefore, cervical cancer like other HPV-dependent tumors shows no increased tumor mutational burden [30,31,32,33]. On the mRNA level, high PD-L1 expression in cervical cancer is either accompanied by high interferon gamma activity as a sign of an ongoing T-cell response or by low interferon gamma activity. Survival outcomes with low activity of interferon gamma are poor, as PD-L1 expression with low interferon gamma activity is triggered by oncogenesis and not by immune-related pathways as in the case of high interferon gamma activity [34]. Immunohistochemistry does not distinguish between these two types of PD-L1 expression. Furthermore, PD-L1 protein expression is altered by post-translational modifications in about 20% of the cases, showing PD-L1 expression on the RNA level but negative staining in immunohistochemistry. This might explain why PD-L1-negative patients may respond to PD-L1/PD-1 inhibition [34,35]. The recently published Phase III Empower Cervical-1 trial, using the PD-1 blocking antibody cemiplimab, showed a significantly longer survival in the cempilimab receiving group. The effect was independent of PD-L1 expression [36]. We failed to show an impaired survival due to a tumoral PD-L1 expression in our PVC cohort. This might be partially attributable to the above-mentioned mechanism of PD- L1 expression. In this context, it is of note that patient number 19 (Table 1) was stable on pembrolizumab for 31 months despite a primary metastatic adenocarcinoma without DES exposure, despite a low tumor mutational burden and despite a mismatch repair proficiency [18]. Only 3 out of 22 patients (13.6%) in our cohort showed a PD-L1 CPS <1. As many of our PVC patients showed a tumoral and immune cell PD-L1 expression, an immune modulatory microenvironment similar to cervical cancer may be present in PVC, thereby supporting the current concept of therapy adaption to that of cervical cancer and vulvar cancer, especially with regard to immunotherapy. Emerging data for immunotherapy in cervical cancer and in vulvar cancer show significant survival benefits in heavily pretreated patients. In the Keynote 826 trial for advanced or metastatic cervical cancer, where pembrolizumab was added to chemotherapy, the percentage of patients with a PD-L1 CPS of >1 was 88.6% [6]. In the EMPOWER 100 trial, where cemiplimab monotherapy was compared to chemotherapy, the rate was 70.7% for the squamous cell cervical cancer patients [36]. The overall response rate in the Keynote 826 trial was 68.1%, the median duration of response was 18 months, and in 22.7% of the patients a complete response was seen. The Empower 100 Trial showed an overall response rate of 16.4% and a median duration of response of 16.4 months for a cemiplimab mono therapy. This effect was independent of the PD-L1 expression [6,36]. In the previous Keynote 158 trial for the cervical cancer cohort, a phase II trial evaluating pembrolizumab mono therapy in recurrent or metastatic cervical cancer, 83.7% of all patients showed a PD-L1-CPS above one. The overall response rate was 12.2%. Within the group of responders, the median duration of response was not reached [37]. In the Keynote 158 vulvar cancer cohort, 83.2% showed a PDL-1 CPS >1 with an overall response rate of 10.9% for a pembrolizumab mono therapy and a median duration of response of 20.4 months [38]. A recent case series from a phase II basket trial reported a significant response in one out of two squamous cell vaginal cancer patients with positive PD-L1-CPS, while the other patient showed progressive disease despite a positive PD-L1 CPS [10]. In our own cohort, we could show that patient no. 19 had a durable response to pembrolizumab for 31 months by the time of this writing [18].

Considering the practice-changing data from the GOG 240 trial in advanced cervical cancer with the implementation of bevacizumab in addition to chemotherapy, this therapeutic target may be a therapeutic option in advanced PVC as well [7,39], especially considering that HPV 16 was the most common identified virus DNA in cervical, vulvar and vaginal cancer—also indicating a similar carcinogenesis in both cancers [40]. Patient no. 13 (Table 1) in our cohort was on 46 months of bevacizumab therapy after her second recurrence within the ileocecal region without any evidence of disease at the time of this writing. In patient no. 19, bevacizumab was not as beneficial, as she recurred after three cycles of bevacizumab monotherapy, after a first response to six cycles of carboplatin/paclitaxel/bevacizumab [18].

P53 mutations are frequently identified across various tumor entities. In the context of HPV positive tumors, the oncogene E6 leads to a degradation of p53 [41]. In HPV negative tumors, p53 may often be mutated. In HPV negative vulvar cancer, a p53 mutation seems to be associated with a disease etiology based on lichen sclerosus [42,43,44]. Data regarding the prognostic impact of p53 mutations in PVC are conflicting [44]. No conclusions regarding prognostic impact or a different carcinogenesis pathway can be drawn so far [45]. In our cohort, p53 failed to show any significance regarding OS and DFS.

There are limitations of our study that have to be mentioned: First, this is a retrospective study design with a small sample size of only 22 patients. On the other hand, we present different therapeutic options in a rare disease, where treatment recommendations, especially for the recurrent or metastatic disease stage, are lacking. Further, we found a high prevalence of p16 expression in PVC which was associated with improved survival. This is in line with previous reports on other HPV-dependent tumors. Furthermore, we documented a high prevalence of tumoral and immune cell PD-L1 expression postulating a similar immune modulatory effect in PVCs as in cervical cancer. As this disease is too rare as to conduct large, randomized trials, therapeutic standards known from cervical cancer regarding bevacizumab and immunotherapy should be considered in these patients, especially in recurrent and advanced disease stages, as this may complement conventional therapy and may improve the situation for these patients with a tumor in one of the most unfavorable locations for surgical treatment and concurrent radiochemotherapy.

## 5. Conclusions

Missing P16 expression, lymph- and hemangiosis and nodal positive disease show decreased survival rates in our cohort of PVC patients. In summary, our data show a similar immunohistochemical expression profile in PVC and similar risk factors for decreased survival rates as in cervical cancer. This suggests a similar carcinogenesis and similar immunomodulatory environments of PVC and cervical cancer. This assumption was successfully utilized in treating two patients of our cohort analogously to cervical cancer. Both are long-term survivors, despite metastatic disease, on bevacizumab and pembrolizumab, respectively.

## Figures and Tables

**Figure 1 cancers-15-01046-f001:**
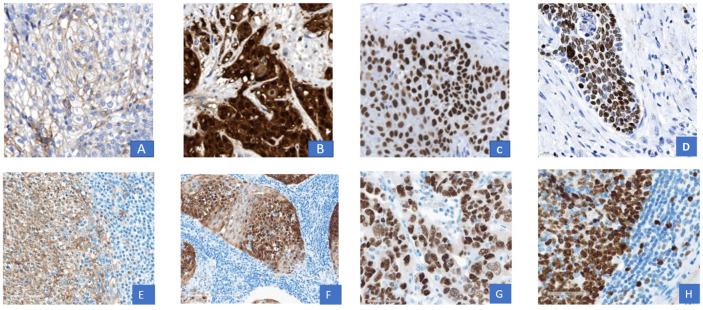
Representative immunohistochemistry for tumoral PD-L1, P16, p53 and KI-67. Representative histology sections show a membranous tumoral PD-L1 expression (**A**), a positive nuclear and cytoplasmatic p 16 staining (**B**), a positive p53 tumor cell nuclei overexpression (**C**) and a strong KI-67 nuclear staining by immunohistochemistry (**D**); validated control sections according to ISO 17020: membranous PD-L1 expression (**E**), nuclear and cytoplasmatic staining of p 16 (**F**), p 53 nuclear overexpression (**G**), KI-67 nuclear and cytoplasmatic staining (**H**) by immunohistochemistry. Magnification 400×.

**Table 1 cancers-15-01046-t001:** Therapeutic course, immunohistochemistry, pathology, and FIGO-stage.

Pt	Age	1st Line	2nd Line	3rd Line	4th Line	5th Line	Rec 1. No 2. Yes	OS (mo)	Survival 1. Dead 2. Alive	FIGO	KI-67 1. >60% 2. ≤60%	PDL1 +≥1 −<1	p53 1. wt 2. mut	P16 +/−	Pathology
1	56	CC/PL					1	13	2	II	1	+	1	+	KSCC
2	61	CC/PL					1	25	2	I	1	+	1	+	NKSCC
3	62	PC/PL					1	26	2	I	2	+	2	+	KSCC
4	87	PC/SEN					1	24	2	I	1	+	1	+	NKSCC
5	64	RHPC/PL					1	48	2	I	2	+	1	+	NKSCC
6	81	PS					1	3	1	IV	2	+	1	+	KSCC
7	47	PE					1	156	2	II	1	−	2	+	NKSCC
8	53	PE					1	108	2	IV	2	+	1	+	ASCC
9	72	PE					1	4	2	IV	2	+	1	−	KSCC
10	66	PE					1	12	2	II	2	+	1	−	CCAC
11	87	PE					1	17	2	II	2	−	1	+	KSCC
12	78	PE					1	4	1	II	2	+	2	−	KSCC
13		PE	IL->RT	HC-> GOG240			2	68	2	II	2	+	1	+	KSCC
14	71	PE	CARB				2	14	1	IV	2	+	1	+	NKSCC
15	68	PE	IL				2	12	1	III	2	−	2	−	NKSCC
16	65	PE					2	9	1	IV	2	+	2	+	NKSCC
17	72	CC/PL					2	96		I	2	+	1	+	NKSCC
18	32	RHPC/PL ->CCRT	CIS/PAC	PE			2	17	1	II	1	+	1	+	KSCC
19	51	GOG240	CIS	PAC/TRA	TD-M1	PEM/RT	2	48	2	IV	2	+	2	+	CCAC
20	75	CCRT					2	7	1	II	1	+	2	−	NKSCC
21	52	CCRT	TACP				2	18	1	III	2	+	2	+	KSCC
22	49	CCRT	PE	RT	PS		2	38	1	III	2	+	2	+	KSCC

Pt: patient; Rec: recurrence; OS: survival in months; PDL1: PD-L1 CPS score; PC/SEN: partial colpectomy + sentinel lymph node dissection; PE: pelvic exenteration; RT: radiotherapy, CCRT: concurrent chemoradiotherapy; RHPC/PL: radical hysterectomy + partial colpectomy + pelvic lymph node dissection; PC/PLNE: partial colpectomy + pelvic lymph node dissection; CC/PLNE: complete colpectomy + pelvic lymph node dissection; IL: ingunial lymph node dissection; GOG 240: 6× carboplatin/paclitaxel/bevacizumab-> becavizumab; CIS: cisplatin; CARB: carboplatin; PAC/TRA: paclitaxel weekly + trastuzumab; PEM: pembrolizumab; PS: palliative surgery (entero- and urostoma); HC: hemicolectomy; TACP: transarterial chemoperfusion with gemcitabine; KSCC: keratinizing squamous cell cancer; NKSCC: non keratinizing squamous cell cancer; ASCC: adenosquamous cell cancer; CCAC: clear cell adenocarcinoma; + deceased.

**Table 2 cancers-15-01046-t002:** 5 year-disease-free survival (DFS) and overall survival (OS).

Parameter	N/%	5-Year DFS	5-Year OS
FIGO	N = 22		
I	5/22.7%	100%	100.00%
II	8/36.4%	50%	56.25%
III	3/13.6%	0%	0%
IV	6/27.3%	20.83%	41.67%
Log-rank:		*p*-value: 0.223	*p*-value: 0.147
N	N = 21 ^+^		
N0	11/50%	81.22%	90.91%
N1	10/45.5%	11.43%	22.86%
Log-Rank		*p*-value: 0.004	*p*-value: 0.004
L	N = 21 *		
L0	14/63.6%	71.43%	68.57%
L1	7/31.8%	0%	17.14%
Log-Rank		*p*-value: <0.001	*p*-value: 0.009
H	N = 21 *		
H0	17/77.3%	63.103%	67.97%
H1	4/18.2%	0%	0%
Log-rank		*p*-value: 0.003	*p*-value: 0.002
Tumor location	N = 22		
Upper third	6/27.3%	50.00%	66.67%
Other Location	16/72.7%	50.91%	47.40%
Log rank		*p*-value: 0.846	*p*-value: 0.482
Tumor size	N = 22		
</=4 cm	11/50%	60%	68.57%
>4 cm	11/50%	36.36%	33.94%
Log-Rank		*p*-value: 0.249	*p*-value: 0.141
</=2 cm	4/18.2%	75%	75%
>2 cm	18/81.8%	39.68%	36.77%
Log-Rank		*p*-value: 0.406	*p*-value: 0.322
T	N = 22		
T1	7/31.8%	71.43%	71.43%
T2	10/45.5%	33.33%	35.56%
T3/T4	5/22.7%	26.67%	40.00%
Log-Rank		*p*-value: 0.627	*p*-value:0.624
R	N = 19 ^#^		
R0	13/59.1%	75.52%	73.43%
R1	6/27.3%	25.00%	50.00%
Log-Rank		*p*-value:0.116	*p*-value: 0.466
Age	N = 22		
</=64 years	11/50%	53.03%	64%
>64 years	11/50%	40.91%	32.91%
Log-Rank		*p*-value: 0.221	*p*-value: 0.029
Depth of infiltration in mm	N = 20 ^−^		
</=5	8/36.5%	75%	58.33%
>5	12/54.5%	37.04%	51.33%
Log-rank		*p*-value: 0.043	*p*-value: 0.248
P16	N = 22		
negative	5/22.7%	26.67%	26.67%
Positive	17/77.3%	52.29%	57.30%
Log-rank		*p*-value:0.230	*p*-value: 0.010
P53	N = 22		
Wild type	13/59.1%	67.69%	71.80%
mutated	9/40.9%	22.22%	22.22%
Log rank		*p*-value: 0.164	*p*-value: 0.091
KI-67	N = 22		
<60%	15/68.2%	42.42%	50.05%
>/=60%	7/31.8%	57.14%	53.57%
Log rank		*p*-value: 0.541	*p*-value: 0.814

^+^: 1 patient had received no lymph node resection due to radiochemotherapy only; *: in 1 patient the biopsy specimen was too small for a true lymph/hemangiosis evaluation; ^#^: 3 patients had received primary radiochemotherapy; ^−^: in 2 out of 3 patients receiving radiochemotherapy, depth of infiltration could not be specified in histology; N: lymph node status; L: lymphangiosis; H: hemangiosis; R: resection margin; T: tumor according to TNM; DFS: disease-free survival; OS: overall survival.

**Table 3 cancers-15-01046-t003:** Multivariate analysis of the prognostic impact of age, lymph node metastases, hemangiosis, lymphangiosis and p16+ staining.

Factor	*p*-Value:
Age	0.365
Lymph node metastasis	0.555
Hemangiosis	0.953
Lymphangiosis	0.947
P 16	0.920

**Table 4 cancers-15-01046-t004:** Correlation of clinicopathological factors with immunohistochemistry.

Factor	PD-L1-CPS > 1	Ki67	P53	P16
FIGO I versus FIGO II-IV	*p*-value: 1.000	*p*-value: 0.2743	*p*-value: 1.0000	*p*-value: 0.2899
Tumor location in the cranial Vagina versus other locations	*p*-value: 1.000	*p*-value: 0.1207	*p*-value: 0.3330	*p*-value: 1.0000
Tumor size ≤4/>4 cm	*p*-value: 0.0152	*p*-value: 0.1984	*p*-value: 0.0805	*p*-value: 1.0000
Depth of infiltration ≤5/>5 mm	*p*-value: 1.000	*p*-value: 0.6126	*p*-value:1.000	*p*-value: 1.0000
Lymph node negative versus lymph node metastasis	*p*-value: 0.5765	*p*-value: 1.0000	*p*-value: 0.0805	*p*-value: 1.0000
L0 versus L1	*p*-value: 1.000	*p*-value: 0.6384	*p*-value: 0.0260	*p*-value: 1.0000
V0 versus V1	*p*-value: 0.465	*p*-value: 0.5743	*p*-value: 0.0096	*p*-value: 0.2098

L0: no lymphangiosis, L1: Lymphangiosis, V0: no hemangiosis, V1: hemangiosis.

**Table 5 cancers-15-01046-t005:** PD-L1 Immunohistochemistry.

PD-L1-CPS	N/%	DFS	OS
</=5	7	71.43%	85, 71%
>5	15	34.91%	36.11%
Log-rank		*p*-value: 0.110	*p*-value: 0.116
PD-L1-TPS			
<10	11	60%	60%
>/=10	11	30.49%	41.56%
Log-rank		*p*-value: 0.110	*p*-value:0.205
PD-L1-IC			
</=1	12	48.61%	54.69%
>1	10	45.71%	49.22%
Log-rank		*p*-value: 0.650	*p*-value: 0.923

## Data Availability

All Data generated and analyzed in this study are included in this article. Further enquiries can be directed to the corresponding author.

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
