# Peer review of "The Role of P16, P53, KI-67 and PD-L1 Immunostaining in Primary Vaginal Cancer"

_cancers, 2023, doi:10.3390/cancers15041046_

Round 1

Reviewer 1 Report

The manuscript describes the clinical and immunohistochemical correlates of survival in retrospective cohort of vaginal cancer patients. After reading of the manuscript I have some remarks:

1/ The references in the text should be placed in square brackets

2/ The reference list should be changed according to the demands of the "Cancers" (see instructions for authors)

3/ Does multivariate analysis also support the conclusions ?

4/ Why there were not survivors in the group of patients with FIGO III?

5/ The abbreviations used in the tables should be described below the table.

Author Response

Dear Reviewer, thanks for the helpful remarks:

1/ The references in the text should be placed in square brackets

Changed

2/ The reference list should be changed according to the demands of the "Cancers" (see instructions for authors)

Changed

3/ Does multivariate analysis also support the conclusions ?

No, unfortunately not, we included a multivariate analysis, as seen in table 3 now

4/ Why there were not survivors in the group of patients with FIGO III?

There were 3 Patients with a FIGO stage III. FIGO stage III was set at first diagnosis.

One patient died due to a severe infection after pallaitive chemoperfusion

One patient had massive bleeding from the tumor for which she received an arterial embolisation, after that she declined any further therapy and died a few month later.

One patient developed an inguinal reccurrence and pulmonal metastases and finally died from that. It is included in the text.

5/ The abbreviations used in the tables should be described below the table.

Is inlcuded now.

Reviewer 2 Report

The manuscript "Clinicopathological Correlates of P16, P53 and PD-L1 Immunostaining in Primary Vaginal Cancer" aims to study clinical, pathological and immunohistochemical features and correlate them with survival in Primary Vaginal Cancer.  The manuscript is interesting and the theme is innovative since the absent of pratical guidelines for vaginal cancer treatment.

However, these are significat questions to be addressed:

- The title is not well written. The word "correlates" is a verb and implies as association, or with clinicapthologica features ou OS. Please rewrite the title in order to improve the idea of the manuscript.

- The introduction requires an update. Information should be provided about the general topic in the light of the current literature. The current state of the research field should be carefully reviewed. Please, explore the introduction. 

- In matherials and methods, the authors do not provide the inclusion and exclusion criteria for the patient selection.

- To perform the immunohistochemistry, the authors use TMA for 17 patients and whole-tumor section for 5 ptients. How many cores from each patient dwere used for TMA construction? Please clarify due to tumor hetreogeneity questions.

- The authors do not clarify the TMA construction.

- The subtitle TMA construction should be updated to TMA construction and Immunohistochemistry

- The antibodies the authors do not mention in the results, should not be refered in material and methods: CD68, CD3 and CD20.

- Please clarify the immunohistochemistry procedure (not in detail, but al least the detection system).

- The results regarding the patients and treatments are very descriptive and confusing, Maybe some categories could be created in order to stratify the patients and to correlate with other clinicopathological features. Please clrify the black star in figure 1. 

-In table 1, it is important to understand the number of patients in each parameter. It is commom to have n(%) in the tables. It is more informative and allow to understand how many patients are associated with each parameter. 

- It could be important to associate the immunohistochemistry expression of the selected biomarkers: P16, P53, Ki67 and PD-L1 with some clinicopathological features, such as FIGO system, tumor location and others. Please, separate the characteristics of table 1 in order to analyse the IHQ expression with the clinicoapthological features.  Please provide a new table.

- The authors must present IHQ sections for all the markers. In figure 2, Ki67 is missing and the controls are missing too.

- The discuusion needs an update. Results tthat are not mentioned in the results (VEGR expression) shoul not be disccussed as important in the manuscript. The authors do not highligh the results and the discussion demonstrates this issue. The disccusion does not properly discuss the results obtained.

- The conclusions could be more elaborated and do not respond to the objectives identified in the abstract.

Author Response

Dear Reviewer,

Thanks for the helpful remarks. All changes in the text are highlighted in yellow.

The title is not well written. The word "correlates" is a verb and implies as association, or with clinicapthologica features ou OS. Please rewrite the title in order to improve the idea of the manuscript.

Titel is changed

  • The introduction requires an update. Information should be provided about the general topic in the light of the current literature. The current state of the research field should be carefully reviewed. Please, explore the introduction.
  • Introduction is changed
  • In matherials and methods, the authors do not provide the inclusion and exclusion criteria for the patient selection.
  • Inclusion and exxclusion criteria are included
  • To perform the immunohistochemistry, the authors use TMA for 17 patients and whole-tumor section for 5 ptients. How many cores from each patient dwere used for TMA construction? Please clarify due to tumor hetreogeneity questions.
  • 2 Cores, this is now included in the text
  • The authors do not clarify the TMA construction.
  • We clarified the TMA Construction in the text.
  • The subtitle TMA construction should be updated to TMA construction and Immunohistochemistry
  • This is changed in the text
  • The antibodies the authors do not mention in the results, should not be refered in material and methods: CD68, CD3 and CD20.
  • The antibodies were excluded
  • Please clarify the immunohistochemistry procedure (not in detail, but al least the detection system).
  • We clarified the immunohistochemistry procedure and the detection system in the test
  • The results regarding the patients and treatments are very descriptive and confusing, Maybe some categories could be created in order to stratify the patients and to correlate with other clinicopathological features. Please clrify the black star in figure 1.
  • a new table is included instead of a diagramm, we hope this makes therapeutic courses better visible

-In table 1, it is important to understand the number of patients in each parameter. It is commom to have n(%) in the tables. It is more informative and allow to understand how many patients are associated with each parameter. 

n/% ist included now for each paramter

  • It could be important to associate the immunohistochemistry expression of the selected biomarkers: P16, P53, Ki67 and PD-L1 with some clinicopathological features, such as FIGO system, tumor location and others. Please, separate the characteristics of table 1 in order to analyse the IHQ expression with the clinicoapthological features.  Please provide a new table.
  • New table is included now with immunhistochemical parameters and clinicopathological features . table 4
  • The authors must present IHQ sections for all the markers. In figure 2, Ki67 is missing and the controls are missing too.
  • A new Figure is included with all IHQ sections and with the controls
  • The discuusion needs an update. Results tthat are not mentioned in the results (VEGR expression) shoul not be disccussed as important in the manuscript. The authors do not highligh the results and the discussion demonstrates this issue. The disccusion does not properly discuss the results obtained.
  • Discussion ist updated
  • The conclusions could be more elaborated and do not respond to the objectives identified in the abstract.
  • Conclusion is now more elaborated and updated.

Reviewer 3 Report

the author presents  an valid analysis of their series xo analyze clinical, pathological and immunohistochemical data.

However a separate table of recurrence is necessary to understand better their analsys.some results of immunohistechemical data needs of a clear table to improve the paper that could be result difficult for interapration of the result. the study is interesting even if the etereogeneuos treatment  could lead to misinterpretation of the final result.I raccomend a more clear exsplanation of the data. A dermographic table also could be useful and of the perfomred treatment. histological type also need to be specified.

Author Response

However a separate table of recurrence is necessary to understand better their analsys.

DFS and OS ist analyzed in Table I for each factor and number of patients are included now. a further table with all therapuetic courses and the immunhistochemistry and the FIGO stage  and histology is included. Patients marked in light grey are the ones with a reccurrence. Table I

some results of immunohistechemical data needs of a clear table to improve the paper that could be result difficult for interapration of the result.

->We correlated all immunhistochemical data to clinicopathological data as seeen in table 4

I raccomend a more clear exsplanation of the data.

->We included in Table 1 the numbers of Patients affected for each parameter

_>Also we included the correlation of clinicopathological parameters with the Immunohistochemistry as depicted in Table 4

->Furthermore we included a multivariate analysis of all significant factors as depicted in table 2

A dermographic table also could be useful and of the perfomred treatment.

Table I is newliy included and shows the therapeutic courses, the FIGO stage, the  immunohistochemistry and histology

histological type also need to be specified.

In table I we included the histological type for each patient

Round 2

Reviewer 1 Report

The head of the table 1 needs editing. Maybe horizontal view will help.

The references are still not in the style of Cancers. The year of publication should be in bold, the title of the journal and its number in italics.

Author Response

The head of the table 1 needs editing. Maybe horizontal view will help. 

Head of the Table 1 ist edited, View ist horizontal now and everything is on one page. 

The references are still not in the style of Cancers. The year of publication should be in bold, the title of the journal and its number in italics.

Year of publication is in bold and Titel and Numer is in italics.